# Systematic Approaches to Study Eclipsed Targeting of Proteins Uncover a New Family of Mitochondrial Proteins

**DOI:** 10.3390/cells12111550

**Published:** 2023-06-05

**Authors:** Maayan Mark, Ofir Klein, Yu Zhang, Koyeli Das, Adi Elbaz, Reut Noa Hazan, Michal Lichtenstein, Norbert Lehming, Maya Schuldiner, Ophry Pines

**Affiliations:** 1Department of Molecular Genetics and Microbiology, IMRIC, Faculty of Medicine, Hebrew University, Jerusalem 9112102, Israel; maayanm@ekmd.huji.ac.il (M.M.); koyeli.das@mail.huji.ac.il (K.D.); adi.elbaz1@mail.huji.ac.il (A.E.); reut.hazan@mail.huji.ac.il (R.N.H.); 2Department of Molecular Genetics, Weizmann Institute of Science, Rehovot 7610001, Israel; ofir.klein@weizmann.ac.il (O.K.); maya.schuldiner@weizmann.ac.il (M.S.); 3CREATE-NUS-HUJ Program and Department of Microbiology and Immunology, Yong Loo Lin School of Medicine, National University of Singapore, Singapore 138602, Singapore; e0335742@u.nus.edu (Y.Z.); micln@nus.edu.sg (N.L.); 4Department of Biochemistry and Molecular Biology, IMRIC, Faculty of Medicine, Hebrew University, Jerusalem 9112102, Israel; michallic@ekmd.huji.ac.il

**Keywords:** mitochondria, eclipsed protein targeting, yeast model system, TDH2, TDH3, dual protein targeting

## Abstract

Dual localization or dual targeting refers to the phenomenon by which identical, or almost identical, proteins are localized to two (or more) separate compartments of the cell. From previous work in the field, we had estimated that a third of the mitochondrial proteome is dual-targeted to extra-mitochondrial locations and suggested that this abundant dual targeting presents an evolutionary advantage. Here, we set out to study how many additional proteins whose main activity is outside mitochondria are also localized, albeit at low levels, to mitochondria (eclipsed). To do this, we employed two complementary approaches utilizing the α-complementation assay in yeast to uncover the extent of such an eclipsed distribution: one systematic and unbiased and the other based on mitochondrial targeting signal (MTS) predictions. Using these approaches, we suggest 280 new eclipsed distributed protein candidates. Interestingly, these proteins are enriched for distinctive properties compared to their exclusively mitochondrial-targeted counterparts. We focus on one unexpected eclipsed protein family of the Triose-phosphate DeHydrogenases (TDH) and prove that, indeed, their eclipsed distribution in mitochondria is important for mitochondrial activity. Our work provides a paradigm of deliberate eclipsed mitochondrial localization, targeting and function, and should expand our understanding of mitochondrial function in health and disease.

## 1. Introduction

It has become evident in recent years that targeting proteins to multiple locations is more abundant than was initially assumed [1,2,3,4,5,6,7]. This phenomenon is termed dual targeting, dual localization, or dual distribution [3,8,9,10,11,11]. In such cases, the identical, or nearly identical, forms of the proteins that are localized to different subcellular compartments are termed echoforms or echoproteins (to distinguish them from isoforms/isoproteins) [11]. Protein dual targeting can be achieved by a variety of molecular mechanisms (for reviews, see [11,11,12,13,14,15,16]). Briefly, dual-targeting mechanisms can either be the result of multiple translation products (for example, due to varying transcription/ translation initiation or termination sites) or of a single translation product due to several competing targeting signals or one ambiguous signal. “Reverse translocation” is another example, in which a subpopulation of the molecules moves back into the cytosol during the translocation process [2,15,17,18].

Since the precise subcellular localization of a protein is critical for its function, one of the challenges facing post-genomic biology is exploring protein subcellular localization to characterize location-specific functions. Indeed, there is a growing number of studies designed to address this challenge by developing global screens of protein localization [19,20,21,22,23,24,25,26]. These studies have provided considerable information regarding protein localization. However, these systematic analyses, such as visualization of fusion proteins with a fluorophore or by mass-spectrometry, often fail to detect one of the echoproteins due to a highly uneven distribution between compartments. This phenomenon is termed ‘eclipsed distribution’ in which the relatively large amount of an echoprotein in one subcellular compartment obscures the detection of the small amount in the other location [3,4] (Figure 1A). Not only the existence but also the function of one echoprotein may be eclipsed by that of the other [3,8,27,28]. We maintain that for the above reasons, the eclipsed distribution phenomenon has a wider incidence than is currently recorded (Figure 1B) and will probably be important for understanding mitochondrial dysfunction and disease.

Our previous studies in the model organism *Saccharomyces cerevisiae* (from here on termed simply yeast) employed an α-complementation technology that uses a split β-galactosidase system to determine protein subcellular localization (Figure 2A) [29,30]. Together with bioinformatics analysis, we estimated that a third of the mitochondrial proteome is dual-targeted to other subcellular compartments [31]. In fact, a major discovery was that many well-studied mitochondrial proteins, such as aconitase, were found to be eclipsed in the cytosol [3,28]. Furthermore, we found that these proteins are enriched by distinctive properties such as a total net charge of the whole protein and weaker mitochondrial targeting signals (MTS) [31,32] and that they are more evolutionary conserved than their exclusive mitochondrial counterparts [33].

In this study, we ask the opposite question—whether there are proteins that were known to reside in other cellular areas yet are eclipsed in mitochondria and have failed to be identified by less sensitive methods. We propose that many proteins, which are thought to be non-mitochondrial (e.g., cytosolic, Figure 1B, orange section), are actually located at very low levels (eclipsed) and may function in mitochondria (Figure 1B, brown section).

To test this, we again employed a variation of the α-complementation assay (Figure 2A) [34] for two complementary screens in the model eukaryote Saccharomyces cerevisiae. In the first, we created and screened a genome-wide collection of yeast strains, each harboring one small fragment (α) of the β-galactosidase on their C’ terminus of a particular protein and then assayed their complementation capacity with either mitochondrial or cytosolic large complementing β-galactosidase fragments (ω, Figure 2B) [34,35]. The second was a hypothesis-driven screen in which we took cytosolic proteins that had a high-scoring predicted MTS-like sequence and assayed them for eclipsed distribution in mitochondria. Using both of these approaches, we discovered hundreds of candidates for eclipsed distribution in mitochondria. We chose the Triose-phosphate DeHydrogenases (TDH) protein family as a test case of eclipsed distribution and showed that they are indeed eclipsed. Moreover, we demonstrate that their mitochondrial form has a role in mitochondrial activity. This study changes the perception of protein localization and, specifically, the mitochondrial proteome in eukaryotes.

**Figure 1 cells-12-01550-f001:**
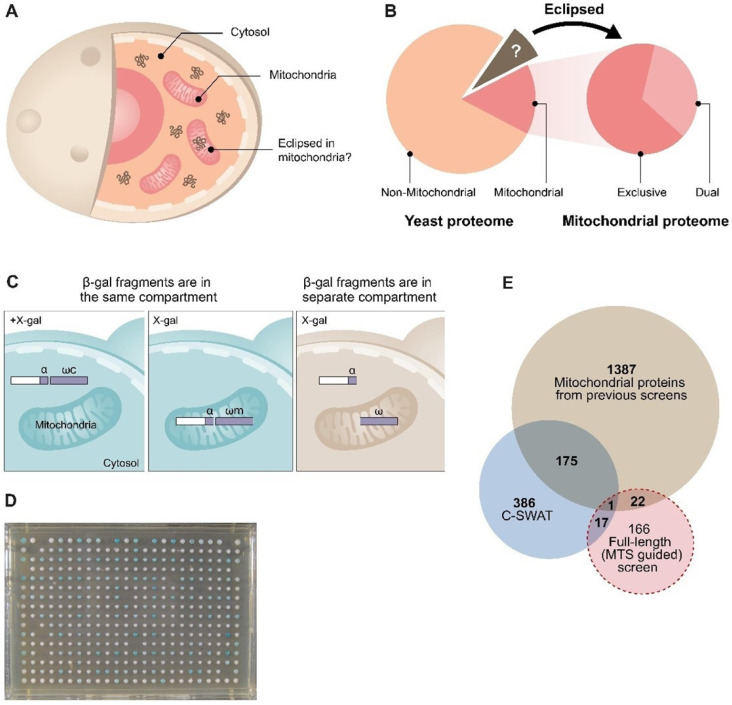
Eclipsed distribution in mitochondria. (**A**) Schematic illustration of eclipsed distributed proteins in mitochondria. The relatively large amount of one echoform in one subcellular compartment (cytosol) obscures the detection of the small amount of the other echoform in the second location (mitochondria). (**B**) A subset of the yeast proteome (left) is localized to mitochondria. A third of the mitochondrial proteome is dual-targeted [31]. The non-mitochondrial proteome is hypothesized to contain a subset of proteins that are eclipsed and distributed in mitochondria (brown section and arrow). (**C**) Schematic illustration of the ɑ-complementation assay that is based on co-localization of β-galactosidase enzyme (β-gal) fragments, ɑ and ω, within the same compartment. β-gal activity can be detected by the formation of blue colonies on X-gal plates (left). In the case that the two fragments are in separate compartments, the colonies appear white (right). (**D**) A representative image of a plate from the C-SWAT library plated on X-gal showing colonies in the 384-well format submitted to the α-complementation assay (the library contains 30 plates). Genes of the library were tagged with α at their C-terminus and mated with a strain expressing either cytosolic or mitochondrial ω. (**E**) Venn diagram of previous studies and both screens from this manuscript. Illustration of the mitochondrial candidate proteins suggested by the C-SWAT (blue sphere, 386 proteins), full-length (MTS-guided) α-complementation screen (red sphere, 166 proteins), and previously alleged mitochondrial proteins (green sphere, 1387 proteins).

**Figure 2 cells-12-01550-f002:**
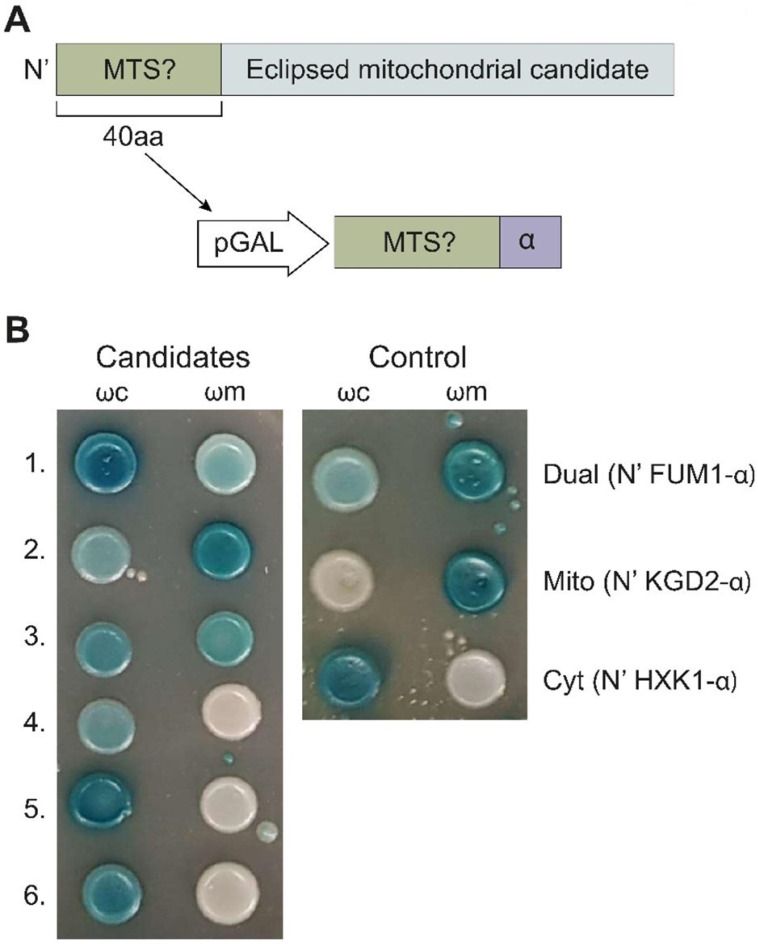
α-complementation analysis of predicted MTS sequences uncovers new candidate echoforms. (**A**) Illustration of the structure of proteins (grey) chosen for analysis of eclipsed mitochondrial distribution. The first 40 amino acids (aa) contain a predicted MTS (green), which was fused to α (purple), expressed under the GAL10 promotor (white), and submitted to the α-complementation assay. (**B**) Example of yeast colonies expressing cytosolic ω (ωc) or mitochondrial ω (ωm) co-expressed with various α-fused N-terminal peptides grown on galactose medium containing X-gal. Blue colonies represent fragments that are associated with the indicated ω fragments. Controls: 40 N-terminal aa of FUM1-α (dual-targeted), KGD2-α (mitochondrial), and HxK-α (cytosolic). For the full list, see Appendix A.

## 2. Results

### 2.1. Genome-Wide Analysis of the Yeast Proteome Reveals 137 Candidates of Eclipsed Mitochondrial Proteins

Building on the assumption that there are unidentified eclipsed mitochondrial proteins (Figure 1B), we employed an α-complementation screen of the whole yeast genome (Figure 1C,D). A genomic SWAp-Tag (SWAT) library of approximately 6000 yeast genes was utilized as a parental library [35]. Using the SWAT approach, we replaced the endogenous tag with one creating in-frame fusions of the α-tag on the C’ of each protein. Since this library contains genes in their native locus, under the regulation of their native promoter, this reduces the risk of false positives due to over- or misexpression. The agar plates with yeast colonies in 384-well format were mated with yeast that express a mitochondrial-matrix targeted β-galactosidase omega fragment (ωm) (Figure 1D) and were scored for their ability to create a blue colony phenotype on X-gal plates, indicating mitochondrial localization.

We detected 386 blue colonies (Appendix A), out of which 176 proteins were previously reported to be mitochondrial (Appendix A, Figure 1E, blue sphere). We did not uncover all mitochondrial proteins since our assay only expects to detect matrix proteins, and many matrix proteins may not be expressed under the conditions assayed or may be unstable when fused to the α tag. We found that 116 of our hits are membrane proteins according to the gene annotation (GO) database (Appendix A). Since complementation demands that their termini face the matrix, these are suggested to be inner membrane proteins whose C terminus faces the matrix. Moreover, 73 of these were never suggested to be mitochondrial, and this presents them as new candidates for inner membrane proteins. Of the remaining soluble proteins, we suggest that 137 are candidates of eclipsed mitochondrial matrix proteins (Appendix A). Interestingly, GO analysis of the biological process and molecular mechanisms of the full list of 386 blue colonies shows enrichment for metabolic and catabolic processes, but when applying the analysis to only the 137 candidates for eclipsed mitochondrial proteins, there is a marked enrichment for cytosolic protein translation processes (Appendix A).

To verify these results, we randomly chose 20 genes that were positive and 11 that were negative according to the screen (Appendix A) and cloned them into an overexpression α-complementation vector for validation. When overexpressed, all 20 positive strains exhibited a blue phenotype with both ωm (like the positive control Kgd1) and ω expressed in the cytosol (ωc), indicating dual targeting to both mitochondria and the cytosol, which is in full agreement with the C-SWAT screen (Appendix A). Seven out of the 11 genes that were negative in the C-SWAT screen were negative in mitochondria (white colonies with ωm) but positive in the cytosol (blue colonies with ωc), which is in full agreement with the C-SWAT screen. Three genes were negative, both with ωm and ωc, suggesting that these fusion proteins may not be expressed properly from our constructs. Importantly, none of our negative controls gave a mitochondrial signal even when overexpressed, attesting to the accuracy of the screen. In summary, the C-SWAT library screen detected a significant number of potential eclipsed proteins targeted to mitochondria.

### 2.2. A Proteomics Approach to Uncover Eclipsed Mitochondrial Proteins

As an additional avenue to uncover eclipsed proteins, we took a proteomics approach using iTRAQ (isobaric tags for relative and absolute quantitation) [36] mass spectrometry of mitochondrial versus cytosolic fractions (see thesis [37]). This approach identified mitochondrial, dual-targeted, and cytosolic proteins with reasonable success (71–98%, 72–93%, and 51–73%, coverage of test groups, respectively). It appears to be much less precise than the C-SWAT approach, even though some new proteins were detected as eclipsed candidates. Candidate proteins from this approach were over-expressed and examined by the α-complementation assay, and 12/17 (70%) were verified as having an eclipsed distribution (Appendix A).

### 2.3. α-Complementation Analysis of Predicted MTSs

To complement our above efforts and create a list of eclipsed protein candidates, we initiated a hypothesis-based approach in which we bioinformatically examined for traits associated with mitochondrial proteins as a means to directly identify eclipsed candidates. We considered those proteins that were predicted to have a high-scoring amino terminal MTS by the MitoProt II algorithm [32,33,38]. Our list contained 542 proteins not known to be mitochondrial but predicted to harbor an MTS with a high probability (above 0.55). From this list, we removed proteins suggested to be membrane proteins (according to gene ontology (GO) annotation), although they could be new mitochondrial inner membrane proteins since the downstream determination of their localization is experimentally more complicated. Following this, we remained with 313 potential eclipsed mitochondrial proteins (Appendix A).

We chose the ~200 highest-scoring proteins (MTS score > 0.7, Appendix A) and examined only their first 40 amino acids, which contain the predicted MTS (Figure 2A) for their capacity to drive the α fragment into the mitochondrial matrix (Figure 2B, Appendix A). A-tagged MTSs whose analysis resulted in blue colonies both with ωc and ωm (BB, Appendix A) were considered dual-targeted, while those that led to blue colonies only with ωc and white colonies with ωm (BW, Appendix A) were considered not mitochondrial. More than 50 percent (103 out of 201) of the MTSs displayed the capacity to target proteins into the matrix (Appendix A). Together, these results suggest the existence of many functional MTSs on proteins previously not identified as mitochondrial.

### 2.4. Analysis of Full-Length Proteins Harboring an MTS by α-Complementation

The fact that a sequence can act as an MTS in isolation does not necessarily mean that it does act as an MTS when part of an entire polypeptide chain. Hence in parallel, we took all 313 soluble proteins with predicted MTS-like sequences (MitoProt II score > 0.55, Appendix A) and cloned the full-length protein ORFs fused to the α fragment. Of the 277 that we successfully cloned and sequenced (194 out of which overlap with the previous analysis (MitoProt II score > 0.7, Appendix A)). We found that a significant percentage, 74% (205 out of 277) of these proteins, appear to be dual targeted (BB + BLB) to mitochondria according to the α-complementation screen, including 20 ribosomal proteins (Appendix A). There are a few possible explanations for the rare WW phenotype (7/277, Appendix A) as previously discussed [31]: (i) these proteins are not expressed; (ii) they are membrane proteins (e.g., OCH1, described below); (iii) they are buried in multi-subunit protein complexes, such as mitochondrial ribosomes, which in turn may hinder their assembly into active α-ω complexes. Worth mentioning is that we were also capable of detecting predicted MTS (previous section) and full-length protein candidates fused to α by western blot (Appendix A).

One negative example was examined in more detail. Och1 harbors an MTS-like sequence at its N-terminus (MitoProtII score 0.98, Appendix A). Yet it appeared negative in α-complementation with ωm for both the MTS alone (WW; Appendix A) and the full-length protein fusions (BW; Appendix A). Further experiments [39] show that Och1 is associated with mitochondrial membranes and can be released only by detergent (TX-100) but not with Na_2_CO_3_ and is not sensitive to protease treatment in the absence of Triton. Thus, the N terminus of Och1, even though it has a high MitoProt II score, inserts into the mitochondrial membrane and does not function as an MTS.

To summarize, we detected 208 (Appendix A) candidates of eclipsed mitochondrial proteins, most of which 166 (Appendix A) are clearly targeted to the organelle. Out of these, 23 proteins were reported only in the last few years as mitochondrial candidates (Appendix A). Therefore, our work brings forward 143 (166 minus 23) new mitochondrial matrix candidates. Together our two independent screens, the C-SWAT (spanning 95% of the entire yeast proteome), which is based on the native endogenous expression levels of proteins, and the MTS-guided screen, which employed over-expression, identify 137 (Appendix A) and 143 (Appendix A) novel mitochondrial candidates, respectively.

### 2.5. Bioinformatics Analysis of the Candidate Eclipsed Mitochondrial Proteins

Previously we have reported that dual-targeted mitochondrial proteins are enriched for specific traits, such as lower whole-protein net charge, weaker MTS strength, higher transcript levels, and higher evolutionary conservation when compared to exclusively targeted mitochondrial proteins or non-mitochondrial proteins [31,32,33]. Conservation parameters assayed were low Dn/Ds (ratio of nonsynonymous to synonymous mutations), high CAI (Codon Adaptation Index), a high number of homologs, and a low PGL (propensity for gene loss). We previously hypothesized that such proteins are more evolutionary conserved than exclusive mitochondrial proteins due to separate selective pressures on the proteins’ presence in the different compartments [33].

To test whether our newly found eclipsed proteins also follow these rules, we analyzed the 137 genes found in the C-SWAT screen. Indeed, we found them to be significantly more conserved versus known, exclusive mitochondrial and non-mitochondrial proteins. This is true for conservation parameters: CAI; Dn-Ds; Homologs; and PGL (*p*-Value ≤ 0.001, Appendix A). Another parameter that tends to be high for dual-targeted and dual-functional proteins [33] is higher expression level which is correlated with higher evolutionary conservation. In fact, when analyzing a previously performed quantification analysis using a different C’ tagged fusion with Green Fluorescent Protein (GFP) [40], we see that, indeed, the expression levels of the C-SWAT eclipsed candidates in mitochondria are significantly higher when compared to exclusive, dual and non-mitochondrial proteins suggesting higher conservation (*p*-Value ≤ 0.001, Appendix A) [33,41,42].

With regard to the net charge on the whole protein sequence, the new C-SWAT candidates appear to be most similar to non-mitochondrial proteins and significantly different from the dual and exclusive mitochondrial proteins (*p*-Value < 0.001, Appendix A). The reason for this is not clear but could be the basis of why these proteins are distributed in an eclipsed fashion and were not identified earlier. Using the same rationale, it may not be surprising that parameters referring to the strength of a possible N-terminal MTS (hydrophobic moment (µHd) and basic residues in the N-terminus) are low for these proteins (Appendix A).

We next repeated these analyses for the 166 mitochondrial eclipsed candidates from the MTS-guided screen. Since this screen is based on the candidates containing an MTS-like sequence at their N termini, it is not surprising that these proteins have the highest average µHD and positive charges within the N-terminus and, in this regard, are most similar to the dual-targeted proteins. Of importance is that the net charge on the whole protein is most similar to known exclusive and dual-targeted mitochondrial proteins and most significantly different from non-mitochondrial proteins (*p*-Value < 0.001, Appendix A). We found insignificant changes in transcript level. The conservation parameters (number of homologs and PGL) were similar to known, exclusive mitochondrial and dual-targeted proteins, which supports our hypothesis that they are eclipsed dual-targeted proteins.

### 2.6. TDH Genes Are Eclipsed Mitochondrial Proteins

Our screens brought forward an extensive array of proteins, many with known cellular activities, that could be functional inside mitochondria. We chose to focus on one example to verify our approach and demonstrate the importance of understanding eclipsed distribution for proteins. To this end, we chose the evolutionary conserved *S. cerevisiae* Triose-phosphate DeHydrogenase (TDH) enzymes. TDH enzymes constitute a protein family of three unlinked, highly homologous genes, *TDH1*, *TDH2*, and *TDH3* [43,44,45,46], that encode 332 amino acid polypeptides. *TDH1* is 84% identical to *TDH2* or *TDH3*, whereas *TDH2* and *TDH3* are 95% identical by protein sequence [45,46]. None of the three genes on its own is essential for cell viability; however, the presence of at least one functional *TDH2* or *TDH3* gene has been reported to be essential for cell viability [45]. TDH proteins are amongst the most abundant soluble enzymes in the cytosol [45,46,47], displaying diverse activities in different subcellular locations [48,49,50,51,52,53]. Their most well-characterized roles are in glycolysis, converting glyceraldehyde-3-phosphate (GAP) to 1, 3 bisphosphoglycerate (BPG) in the presence of the cofactor nicotinamide adenine dinucleotide (NAD+) to yield NADH [54]. The N-terminal protein sequence of Tdh3 analysis by the UniProt database, based on predicted structures [55] and MitoProtII [38], suggest the existence of an MTS at the amino terminus (Figure 3A,B and Appendix A). Moreover, sequence analysis of the first 30 amino acids of the Tdh proteins reveals distinct conserved domains across the paralogs throughout eukarya, with a total of 43% amino acid conserved in over 90% of species (Figure 3C). We chose this family since Tdh2 and Tdh3 were positive in both the MTS guided and C-SWAT screens, and they are well-studied and have known functions. While several other systematic manuscripts have found them to be associated with mitochondria [11,22,56], they have never been shown to reside in the mitochondrial matrix.

We examined the distribution pattern of Tdh1, Tdh2, and Tdh3 by subcellular fractionation followed by a western blot with a polyclonal antibody, which recognizes all three wild-type Tdh proteins (Figure 3D). Using this approach, we found that Tdh1/Tdh2/Tdh3 expressed from their natural promoters on the chromosomes collectively exhibit a classical eclipsed distribution with minute amounts in mitochondria (Figure 3D, M vs. C). We also examined the distribution of each of the full-length Tdh proteins when individually overexpressed (Figure 3E) using fusion alleles to a 6His-tag, which allows us to specifically detect each Tdh protein separately. All tagged Tdh proteins, as well as the endogenous (Figure 3D) proteins, exhibited an eclipsed distribution pattern, which means that the majority of Tdh proteins are found in the cytosolic fraction (C), while minute amounts of the proteins are detected in mitochondria (M) (Figure 3E).

### 2.7. Tdh Proteins Rely on MTS-Like Signals for Their Eclipsed Distribution

To determine the reliance on the MTS-like sequence for mitochondrial targeting, we fused either their first 30 amino acids or the full-length Tdh proteins to the α fragment at their C-terminus (Figure 4A). We then followed their localization by α-complementation. Indeed, the MTS alone was sufficient to enable matrix targeting (Figure 4B), similar to the full-length proteins. To test if the MTS is also necessary for matrix targeting, we examined Tdh2 and Tdh3 lacking an MTS (Δ*MTS-TDHα*) and found that they were absent from mitochondria according to the α-complementation assay (Figure 4C and Appendix A). Since these Δ*MTS* constructs appear to lack enzymatic activity, we also created a version that lacks a predicted MTS by amino acid substitutions (mutxp, Figure 4A, Construct 4 illustration). While *mut2p-TDH2α* with only two substitutions of arginines to glutamates (Arg11, Glu14) still had a residual presence in mitochondria, the mutant with five substitutions (*mut5p-TDH2α* in Arg3,11,14,18,23Glu) is already absent from the organelle (Figure 4D and Appendix A). These two mutants, *mut2p-TDH2α*, and *mut5p-TDH2α*, also appear to lack enzymatic activity; hence our experiments cannot clearly distinguish these two traits. Together, these results indicate that both Tdh2 and Tdh3 harbor a functional MTS [55].

### 2.8. Processing of Tdh2/3 Proteins

Tdh proteins contain two potential translation initiation methionine codons; one is the first codon that leads to translation of the MTS, while the other, from codon 17, will lack the MTS. Hence, to determine if the dual targeting of Tdh proteins is a result of two translation products, we created a mutant *mutM17Vα*, in which the 17th amino acid methionine was replaced by valine, removing the potential second start codon (Figure 5A, illustration). This mutant was fused to the α tag and followed by the α-complementation assay to probe localization. We found no visible effect of the mutation on the distribution of Tdh2 and Tdh3 proteins in the cell, suggesting that the dual localization of Tdh2/3 is not due to two translation products (Figure 5B). In support of this, no translation from the second methionine was observed by ribosome profiling [57].

To further examine this question, we fused the whole Tdh2 protein sequence to the strong MTS of fumarase, which is expected to be processed by MPP (*FUMMTS-TDH2α*, Figure 4A). As expected, this construct is processed by MPP, which removes the *FUM-MTS* from some of the molecules, generating two products: A mature (M) species, which displays the same molecular weight as the wild type Tdh2α (Figure 5C, compare lane 4 to lanes 1, 2) and a precursor (P) species which is larger. These two species can be observed more clearly in the presence of the ionophore carbonyl cyanide 3-chlorophenylhydrazone (CCCP), which dissipates the mitochondrial membrane potential and blocks import, resulting in the accumulation of unprocessed proteins in the cytosol (Figure 5C, lane 3). It is important to emphasize that besides these two products, Tdh is not further processed, indicating that the original Tdh2 is not processed. Accordingly, we show by α-complementation that *mut5p-TDH2α* is absent from mitochondria (Figure 4D and Appendix A), but its targeting to mitochondria can be restored by the upstream fusion of *FUM-MTS* (*FUMMTS-mut5p-TDH2α*, Figure 4A Construct 5 illustration and Figure 4D, *α*-complementation). Thus, we conclude that Tdh proteins are not processed, suggesting that the MTS amphiphilic α helix is part of the first functional domain of the protein as predicted by the UniProt database (Figure 3, Appendix A).

### 2.9. Tdh Proteins Are Important for Growth on Carbon Sources Requiring Mitochondria

Since we verified that Tdh proteins indeed have an eclipsed distribution in mitochondria, we wanted to explore their functional role in the organelle. To do this, we constructed a double deletion strain of both *TDH2* and *TDH3* kept viable by a rescue plasmid expressing WT-Tdh2 under an inducible *GAL10* promoter with a counter-selectable URA3 marker [46] (Figure 6A). Use of 5FOA (5 Fluoroorotic acid) can therefore be used to replace plasmids (shuffle) with a plasmid enabling biosynthesis of leucine (LEU) expressing specific variants (Figure 6A).

Growth was examined on an agar medium supplemented with either fermentable (glucose), oxygen-requiring (galactose), or respiration-requiring (ethanol-acetate) carbon/energy sources to examine the different shuffled strains (Figure 6B). All the strains appear to grow readily on a glucose medium, which indicates that the double deletion is viable in our strains and growth conditions (Figure 6B, Row 3). Nevertheless, the double deletion mutant exhibited poor growth in the galactose medium when compared to the wild type or to a single knockout strain, Δ*tdh3* (Figure 6B, Row 3). On ethanol-acetate medium, which requires functional mitochondria, both the double deletion mutants and the single Δ*tdh3* exhibit poor growth (Figure 6B, Rows 2, 3). These galactose and ethanol-acetate phenotypes can be restored by a complementing expression plasmid of *TDH2*ɑ (Figure 6B, Row 4).

We assayed the enzymatic activity of the Tdh protein-shuffled-strains (Figure 6C). The Tdh enzymatic activity of the single mutants Δ*tdh2* or Δ*tdh3* show similarity to that of the wild type. As expected, the double mutant Δ*tdh2*Δ*tdh3* displays significantly lower activity (Figure 6C, bar 4, *p* < 0.05) that can be complemented by Tdh2 expression from a plasmid (Figure 6C bar 5). The background Tdh activity can be attributed to the *TDH1* gene, which exists in all our strains. The Δ*tdh3*-strain activity is slightly lower corresponding to the lower protein expression of Tdh in this strain (Figure 6D lane 3).

### 2.10. Tdh Proteins Affect Mitochondrial Activity When in the Mitochondrial Matrix

Our attempts to isolate mutant variants in which Tdh2/3 are absent from mitochondria but are both present and active outside mitochondria were unsuccessful. As an alternative approach, we expressed the Tdh2/3 proteins solely in mitochondria. To do this, we fused upstream to Tdh3, the sequence of the MTS of subunit 9 of F0-ATPase of *Neurospora crassa* (*Su9-MTS* (1-69 aa) Figure 7A in grey). Su9-MTS is the strongest known MTS and is processed twice by MPP. In addition, we fused the degron tag SL17 (colored blue) to the C terminus of Tdh to target the cytosolic pool of the protein for proteasomal degradation. Expressing *Su9MTS-Tdh3-SL17* creates a hybrid protein that is efficiently targeted to mitochondria, and there, it is protected from degradation while the cytosolic form of the protein (tagged with a degron) is degraded.

When expressing *Su9MTS-Tdh3-SL17* in the absence of CCCP, only the mature processed mitochondrial form of the protein can be detected (~41 KDa), indicating that the cytosolic form has, as expected, been eliminated (Figure 7B). In the presence of CCCP, only the precursor form (~48KDa) can be detected (Figure 7B, lanes 7, 8). Tdh3α (~45 KDa), which lacks an external MTS, as expected, does not exhibit processing (Figure 7B, lanes 3, 4). *Su9MTS-Tdh3α* (~52KDa), which harbors an *Su9MTS*, displays two molecular bands in the absence of CCCP (precursor ~52 KDa and mature ~45 KDa), indicating either partial import into mitochondria or partial *Su9-MTS* cleavage (Figure 7B, lane 6). In the presence of CCCP, only the precursor is detected (Figure 7B, lane 5).

The mutant variant *Su9MTS-Tdh3-SL17* cannot rescue the growth phenotype of Δ*tdh2*Δ*tdh3* on galactose media (Figure 7C), demonstrating an essential role under these conditions for Tdh proteins in the cytosol. Growth on ethanol-acetate media, however, which requires mitochondrial respiration, showed a slight but reproducible increase in the growth of Δ*tdh2*Δ*tdh3* expressing *Su9MTS-Tdh3-SL17*, which is localized only to the organelle relative to its controls (Figure 7C rows 2, 3 and 5). This supports the notion that under these conditions, there is a functional role for the eclipsed Tdh proteins in mitochondria.

To examine this in more depth, we analyzed the quantitative respiratory capacity of yeast cultures grown in an ethanol-acetate medium by determining the oxygen consumption rate (OCR). Basal respiration was measured, which was followed by the addition of FCCP, a mitochondrial oxidative phosphorylation uncoupler, which leads to maximal respiration. Then antimycin A (AA), which inhibits complex 3 in the respiratory chain and blocks respiration, was added (Figure 7D). The spare respiration capacity (Maximal minus basal respiration) was calculated (Figure 7E and Appendix A). We found that, surprisingly, the respiratory capacity of Δ*tdh2*Δ*tdh3* (or Δ*tdh2*Δ*tdh3* containing an empty vector) was 3-fold higher compared to the control or a single Δ*tdh2* deletion. We hypothesize that this may be a result of a compensatory mechanism of the cells. Importantly, *pSu9MTS-Tdh3-SL17*, which is efficiently expressed only in mitochondria, completely restored the normal respiration capacity phenotype of the double mutant. A plasmid expressing full-length pTdh3α only partially complemented the Δ*tdh2*Δ*tdh3* strain as expected since it has only a minor fraction targeted to mitochondria and cannot replace the loss of both homologs.

## 3. Discussion

This study changes the perception of the mitochondrial proteome. We put forward hundreds of eclipsed mitochondrial protein candidates due to the complementarity of our approaches. First, the C-SWAT library enabled us to have an unbiased screen of genes under their natural promoter providing limited sensitivity but a wide coverage. From this approach, the 137 candidates are more abundant proteins that are, on average, significantly more conserved than exclusive mitochondrial and non-mitochondrial proteins supporting their identification as eclipsed proteins.

Our complementary approach was more limited in scope but with much higher sensitivity. Choosing proteins based on the presence of an MTS-like sequence at their amino terminus uncovered 143 candidate eclipsed mitochondrial proteins. These exhibit an enrichment of mitochondrial protein properties, such as their whole-protein net charge that is similar to known exclusive and dual-targeted mitochondrial proteins. With regard to conservation parameters, such as the number of homologs and PGL, they are similar to known, exclusive mitochondrial, and dual localized proteins, which follows our interpretation that they are eclipsed dual-targeted proteins.

Our approaches were complementary and allowed us to capture a plethora of hits, as can be seen by the low (~10%) overlap between the candidates raised by each approach (Figure 1E). In other words, we are looking at the tip of the iceberg. This suggests that additional guided screens, based on metabolic/enzymatic activity, protein traits (such as a net charge on the whole protein), internal MTSs [58,59], or evolutionary conservation, may identify additional eclipsed mitochondrial proteins.

In order to prove that the eclipsed proteins are indeed deliberately targeted to mitochondria and have a function there, it is important to go into an in-depth, mechanistic exploration of each example. We chose the *TDH1/2/3* gene family since it appeared on a number of our screens and also in recent studies [20,25,60]. Focusing on Tdh2/3 that came up in both screens, we found that the natural and overexpressed subcellular pattern of Tdh2/3 is of eclipsed distribution (highly abundant in the cytosol with minute amounts in mitochondria). Tdh2/3 harbor an N terminal “MTS-like” sequence, which can function independently of the whole protein but is not cleaved since its removal or site-specific mutagenesis results in loss of enzymatic activity. We could show that the dual targeting is not a result of two translation products and that the N terminus of these proteins is both necessary and sufficient for targeting. Most importantly, while we do not yet understand the exact role of Tdh proteins in specific mitochondrial metabolic pathways, we could show that the matrix form of the proteins affects mitochondrial-related phenotypes (the capacity to grow on respiration requiring medium and the overall respiration rate). Since the rescue by *Su9MTS-TDH3-SL17* is stronger than the rescue by WT-*TDH3*α, our results suggest that the abundance of matrix echoforms is of importance. Our results provide evidence that Tdh proteins are involved in mitochondrial respiration and confirm echoform dual function. While we have chosen here to present the Tdh1/2/3 family as an example of an eclipsed distribution of proteins in mitochondria, this is not the only case that we are pursuing. Recently we have found that ubiquitination occurs in the mitochondrial matrix by eclipsed targeted components of the ubiquitin machinery [37]. Future single-gene studies should reveal many more examples of functional eclipsed distribution of mitochondrial proteins.

Generally, we have come a long way since the “One gene one enzyme (polypeptide) hypothesis” [61]. Today, not only do we understand that a single gene can give rise to multiple protein products, but we also realize that these protein products are not necessarily targeted to a single organelle or subcellular location. Such cases of dual targeting or localization, which we have shown are highly abundant, provide cellular flexibility and tunability. Our work, therefore, not only expands our understanding of mitochondrial function but also changes the way we perceive protein localization in eukaryotic cells. This study provides a cue for the investigation of eclipsed mitochondrial proteins’ functions in higher eukaryotes and involvement in human disorders. Our work provides a paradigm of deliberate eclipsed mitochondrial localization, targeting, and function and should expand our understanding of mitochondrial function in health and disease.

## 4. Materials and Methods

### 4.1. Strains

*Saccharomyces cerevisiae* Yeast strains were provided by Euroscarf; BY4741 (MATa; his3Δ1; leu2Δ0; met15Δ0; ura3Δ0), Δ*tdh2* (BY4741; MATa; his3Δ1; leu2Δ0; met15Δ0; ura3Δ0; YJR009c::kanMX4), and Δ*tdh3* (BY4741; MATa; his3Δ1; leu2Δ0; met15Δ0; ura3Δ0; YGR192c::kanMX4). *S. Cerevisiae* wild type 7A (YPH499, Sc-7A) and TCA cycle (tricarboxylic acid cycle) mutant *7C* (YPH500, Sc-7C) were provided from our lab stock.

Additional null and composite null mutants were generated for this study using the homologous recombination technique. The Δ*tdh2*Δ*tdh3* double null mutant strain was constructed using transformants Δ*tdh3* carrying pGalTdh2Ura(R10) plasmid. Essentially, the cells were made competent using the Frozen-EZ Yeast Transformation II™ (ZYMO RESEARCH), according to the manufacturer’s instructions. A cloNAT cassette with flanking upstream and downstream sequences of the *TDH2* gene was cloned and amplified by PCR using pAV10.KN (AddGene) plasmid as a template using the following primers:

5′-CCAAGAACTTAGTTTCAAATTAAATTCATCACACAAACAAACAAAACAAAgacatggaggcccagaatac-3′ and

5′-AATTATTAATAATAAAAACTAAATCATTAAAGTAACTTAAGGAGTTAAATcagtatagcgaccagcattc-3′.

The amplified cassette was transformed into the *Δtdh3*+pGalTdh2Ura(R10) strain, and yeast were plated on 2% galactose lacking uracil and containing 200 µg/mL cloNAT as a selective medium. After primary selection, mutant colonies were purified by PCR to confirm complete parental gene knockout using the following primers:

5′-tcggagacctgcaatttt-3′ (*TDH2* chromosomal flanking upstream sequence);

5′-ggttgtttatgttcggatgt-3′ (cloNAT TEF promoter sequence, positive control); and

5′-ccagcgtcaatgtgcttt-3′ (*TDH2* mid chromosomal sequence, negative control).

### 4.2. Plasmids

pωc (pYES/M15) was kindly provided by Picard [29], pFumα, pHxk1α, pKgd1α, and BS-Su9ω (pωm) were described elsewhere [34]. All the screened proteins used in this study were created by synthesis and amplification of the corresponding fragments by Bio-Basic Singapore. The resulting products were cloned into pFumα using an orientation enrichment reaction (OER). All plasmids described above were introduced into strain BY4741.

Su9-MTS—MASTRVLASRLASQMAASAKVARPAVRVAQVSKRTIQTGSPLQTLKRTQMTSIVNATTRQ

AFQKRAYSS

SL17—SISFVIRSHASIRMGASNDFFHKLYFTKCLTSVILSKFLIHLLLRSTPRV

FUM-MTS—MLRFTNCSCKTFVKSSYKLNIRRMNSS

p*TDH*(1–3)-6His—Tdh(1–3) were tagged with 6His at their C-terminus and cloned into pRS423Gal. These plasmids were created by amplifying the respective sequences from genomic DNA using the indicated primers below with BamHI and EcoRI at ′5 and ′3, respectively. The amplified PCR fragments were cut with BamHI and EcoRI and cloned into pRS423Gal cut with BamHI and EcoRI.

Primers: For *TDH*1—5′-aaatttggatccATGATCAGAATTGCTATTAACG-3′ and 5′-tttaaagaattcGTGATGGTGATGGTGATGTTAAGCCTTGGCAACATATTC-3′.

For *TDH*(2–3)—5′-aaatttggatccATGGTTAGAGTTGCTATTAACG-3′ and 5′-aaatttgaattcGTGATGGTGATGGTGATGTTAAGCCTTGGCAACGTGTTC-3′.

### 4.3. Growth Conditions

Strains were grown at 30 °C or as indicated in synthetic depleted medium containing 0.67% (*w*/*v*) yeast nitrogen base without amino acids (Difco Laboratories, Franklin Lakes, NJ, USA), 2% glucose or galactose (*w*/*v*), CSM dropout mix (Qbiogene, Solon, OH, USA) supplemented with the appropriate amino acids (50 μg/mL). For agar plates, 2% agar was added. X-gal plates were prepared as follows; a stock of 2% galactose, 1% raffinose, 0.008% X-gal (dissolved in 100% N, Ndimethylformamide), and 1× BU salts (25 mM sodium phosphate buffer titrated to pH 7.0) was added to an autoclaved medium at a temperature of 50 °C. For 5′FOA selection, strains containing pGalTdh2Ura3 (pR10) plasmid were streaked on 2% galactose plates supplemented with 0.1% 5-fluoroorotic acid for selection of the loss of pGalTdh2Ura3. Colonies were picked and streaked on galactose media lacking URA3 to confirm the loss of pR10.

### 4.4. β-Galactosidase α-Complementation Assay

Yeast cells were transformed with plasmids encoding various α fusion proteins and either ωc or ωm. Colonies were plated on X-gal plates and incubated at 30 °C for 72 h.

### 4.5. C-SWAT Library

The C-terminus SWAp Tag (SWAT) library of the yeast Saccharomyces cerevisiae [35] was used to generate a library with a C-terminus tag as previously published (same reference). In short, a SWAT donor strain (yMS2085) was transformed with a donor plasmid (pMS859) containing the alpha tag and then SWATted as described. The final library genotype is *his3*∆*1 leu2*∆*0 met15*∆*0 ura3*∆*0*, *can1*∆*::GAL1pr-SceI-NLS::STE2pr-SpHIS5 lyp1*∆*::STE3pr-LEU2* (XXX::L3-alpha-ADH1ter-Hygro-ALG9term-L4. Once generated, the swatting efficiency and integrity were evaluated (Appendix A). For the α-complementation assay, the library was mated with BY4742 (MATα; *his3*Δ*1*; *leu2*Δ*0*; *lys2*Δ*0*; *ura3*Δ*0*), containing ωm expression plasmid (described above), and plated on X-gal. The plates were incubated at 30 °C for 72 h.

### 4.6. Metabolic Labeling

Yeast cultures were grown in 5 mL SD–glucose or SD–galactose at 30 °C or as indicated to OD_600 nm_ = 1.5. Cells were collected by centrifugation and resuspended in 400 μL fresh media lacking methionine. Cultures were labeled with 40 μCi/mL [^35^S] methionine and further incubated for 30 min at 30 °C. When required, carbonyl cyanide chlorophenylhydrazone (CCCP) was added to a final concentration of 20 μM at the start of labeling. Labeling was stopped by the addition of 10 mM sodium azide. The labeled cells were collected by centrifugation and resuspended in TE buffer (Tris10 mM, EDTA 1 mM, pH 8.0) containing 1 mM phenylmethylsulfonylfluoride (PMSF), broken with glass beads, and centrifuged to obtain the supernatant fraction. Supernatants were denatured by boiling in 1% SDS, immunoprecipitated with the indicated anti-serum and protein A–Sepharose (Amersham Biosciences, Amersham, UK) or magnetic Dynabeads protein A (Dynal Biotech ASA, Oslo, Norway). Samples were then analyzed by SDS-PAGE, followed by visualization with the BAS2000 image analyzing system (Fuji Corp., Zhitachi, Japan) and autoradiography.

### 4.7. Subcellular Fractionation

Induced yeast (in galactose) cultures were grown to OD_600 nm_ = 1.5. Mitochondria were isolated as described previously [18]. Spheroplasts were prepared in the presence of Zymolyase-20T. Subcellular fractionation experiments were assayed for cross-contamination of fractions using Hsp60 as a mitochondrial marker and Hexokinase1 as a cytosolic marker in Western blots.

### 4.8. Western Blot Analysis

*S. cerevisiae* cells were harvested using lysis buffer containing the following: 10 mM Tris pH 8, 1 mM EDTA, and 100 mM PMSF. Protein concentrations were determined using the Bradford method [62]. Samples were separated on 10% or 12% SDS-PAGE gels and transferred onto PVDF membranes (Millipore, Burlington, MA, USA). The indicated primary antibodies were used. All blots were incubated with the appropriate IgG-HRP–conjugated secondary antibody. Protein bands were visualized and developed using an enhanced chemiluminescent immunoblotting detection system (ImageQuant LAS 4000 mini, GE Healthcare, Chicago, IL, USA) and a gel documentation system (Bio-Rad, Hercules, CA, USA).

### 4.9. Enzymatic Activity

*S. cerevisiae* cultures grown in galactose medium for translation induction were harvested and resuspended in TE buffer (pH 8.0) containing 1 mM PMSF, broken with glass beads, and centrifuged to obtain the supernatant fraction. Glyceraldehyde-3-phosphate dehydrogenase activity was assayed in 1.0-mL reactions containing 0.1 M potassium phosphate (pH 7.4), 1.0 mM NAD+, 10 mM EDTA, 0.1 mM dithiothreitol, and 4.0 mM glyceraldehyde 3-phosphate [28]. NADH formation was monitored spectrophotometrically at 340 nm. Glyceraldehyde-3-phosphate dehydrogenase activity units are expressed as µmol of NADH formed/min at 25 °C.

### 4.10. Oxygen Consumption Evaluation by Seahorse

Mitochondrial function was assessed using the XF96 extracellular flux analyzer (Agilent, Santa Clara, CA, USA) as previously described [63], with minor modifications. Yeast cultures were grown overnight at 30 °C in galactose or galactose selected medium. Cells were diluted to OD_600 nm_ = 0.1 in ethanol-acetate medium, and 180 µL were seeded in Seahorse XF96 microplate coated in poly-L-lysine (22 µL of 0.1 mg/mL per well). The loaded plate was centrifuged at 500× *g* for 3 min at room temperature, following incubation for 30 min at 30 °C. Basal Oxygen Consumption Rate (OCR) was measured following an injection of 20 µM of FCCP and 2.5 µM of antimycin A according to the manufacturers’ manual Seahorse XF Cell Mito Stress Test (Agilent). OCR was normalized to cells at OD_600 nm_ = 0.1, and basal respiration, maximal respiration, and spare respiration capacity were calculated according to Agilent protocol.

### 4.11. Microscopic Analysis of Mitochondria Phenotype

Glass-bottom, 384-well microscopy plates (Matrical Bioscience, Spokane, WA, USA) coated with Concanavalin A (Sigma-Aldrich, St. Louis, MI, USA) were used for imaging. Cells in the mid-logarithmic phase were adhered to the plates by incubating at RT for 15 min. Upon adherence to the plate, media was replaced with media containing 50 nM MitoTracker (MitoTracker Orange CMTMRos; Invitrogen, Waltham, MA, USA), and cells were incubated at RT for 10 min, washed once, and imaged using automated inverted fluorescence microscope system (Olympus, Tokyo, Japan) harboring a spinning disk high-resolution module (Yokogawa CSU-W1 SoRa confocal scanner with double microlenses and 50-µm pinholes). Images of cells in the 384-well plates were using a 60× oil lens (NA 1.42) and with a Hamamatsu ORCA-Flash 4.0 camera. Mitottracker was excited by a 561 laser, and the signal was detected through a 617/73 nm emission filter. For each well, 9 positions were imaged, and over 6000 yeast cells were segmented for mitochondria, according to the mitotracker signal, and analyzed using ScanR software (Olympus, version 3.2.0).

### 4.12. iTRAQ Assay

An amount of 100 μL of no more than 110 μg proteins of each fraction sample (use approximately equal amounts of proteins for different fraction samples) with 10 μL of 1 M TEAB and 2 μL of 5% SDS was reduced with 5 μL of 100 mM TCEP and capitalized with 5 μL of 200 mM MMTS. An amount of 20 mL of 150 mM TEAB with 0.05% SDS was used to dialyze each capitalized sample overnight using Tube-O-DIALYZER™, Micro 1K MWCO. An amount of 2–3 μg trypsin was used to trypsinize each dialyzed sample. iTRAQ labeling of trypsinized samples was conducted using a SCIEX kit. The 8 differently labeled samples were then pooled together. The pooled mixture was then subjected to cation exchange chromatography followed by liquid chromatography-tandem mass spectrometry (LC-MS/MS), which was conducted by Protein and Proteomics Centre (PPC), NUS Biological Sciences. Data processing information can be found at https://scholarbank.nus.edu.sg/handle/10635/236754, accessed on 2 May 2023.

### 4.13. Statistical Analysis

All statistical analyses in Appendix A were carried out by GraphPad Prism, version 9.3.1. *p*-value was calculated for unpaired and non-parametric tests using the Mann-Whitney test (to compare ranks) or the Kolmogorov-Smirnov (to compare cumulative distribution). GO enrichment analysis in Appendix A was performed as described in [64], using Fisher’s exact statistical significance test, showing only results of FDR *p* < 0.05.

## Figures and Tables

**Figure 3 cells-12-01550-f003:**
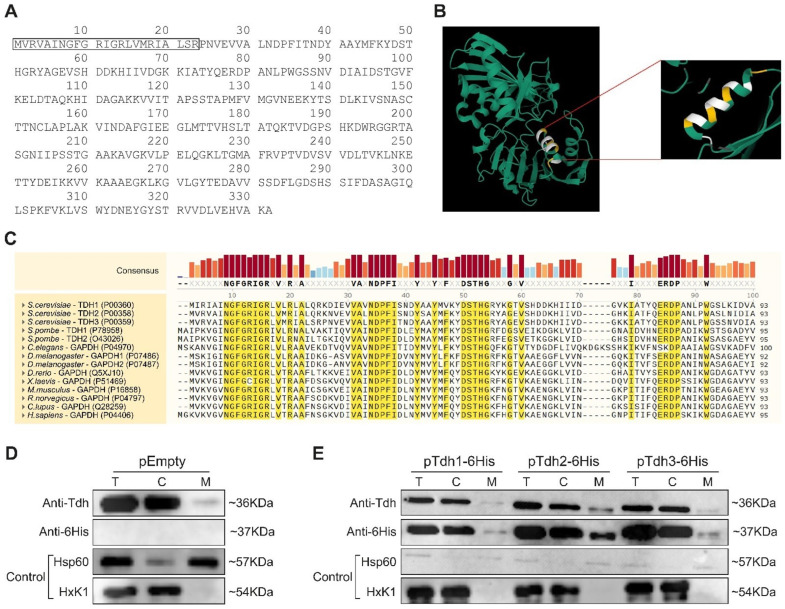
Tdh3 has a potential MTS-like signal and displays an eclipsed distribution pattern in mitochondria. (**A**) Tdh3 amino acid (aa) sequence. The predicted MTS is boxed and enriched with positively charged and hydrophobic aa. (**B**) Tdh3 structure (green) according to the uniport database, https://www.uniprot.org/uniprot/P00359#structure, accessed on 2 May 2023. The box is a zoom-in on the predicted amphiphilic alpha-helix that comprises the MTS (yellow, positively charged aa; white, hydrophobic aa). (**C**) Amino acid sequence alignment between Tdh paralogs across eukaryotic species. Uniport numbers are in brackets. Marked in yellow are amino acids which are conserved in over 90% of sequences. Consensus between sequences for all amino acids is ranked in the upper panel and labeled by bar and color coding. (**D**) Western blot analysis of subcellular fractionation on wild-type yeast cells harboring an Empty pRS423 vector. Equivalent portions of fractions of the total (T), cytosol (C), and mitochondria (M) were analyzed using antibodies against Tdh; 6His; Hsp60 as a mitochondrial marker and HxK1 as a cytosolic marker. (**E**) Wild-type yeast strains harboring the indicated pRS423 plasmids were subjected to subcellular fractionation and analyzed by western blotting as described in (**D**).

**Figure 4 cells-12-01550-f004:**
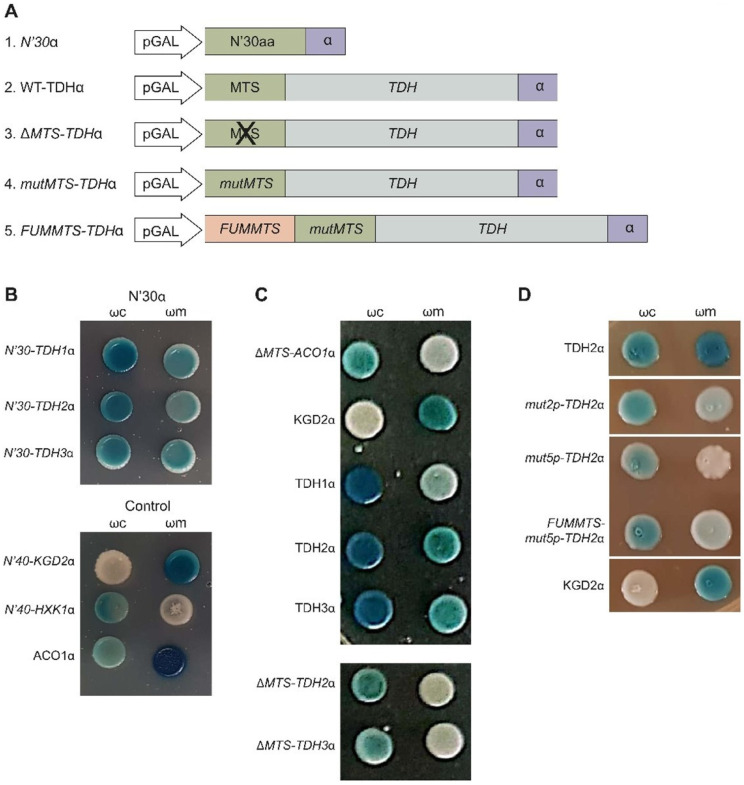
Tdh2/3 are MTS-dependent mitochondrial dual-targeted proteins. (**A**) Illustrations of the *TDH* gene constructs utilized. These *TDH* (grey) constructs contain a predicted MTS (green), expressed under the GAL10 promotor (white), and are fused to an α tag (purple). 1. *N’30α*, N-terminal first 30 amino acids (aa) of Tdh fused to *α*; 2. Tdhα wild-type (WT); 3. Δ*MTS-TDHα*, deletion of the N terminal 16 aa; 4. *mutMTS*-Tdh*α*, point mutations in the predicted MTS; 5. *FUM-MTS* (green), the MTS of fumarase, is fused to the N-terminus of the *TDH* ORF. (**B**) α-complementation of *TDH* predicted MTSs. Yeast colonies expressing cytosolic ω (ωc) or mitochondrial ω (ωm) were co-expressed with the α-fused N-terminal peptides of Tdh1, Tdh2, and Tdh3 (first 30 aa, as described in (**A**), illustration 1). Yeast cultures were grown in galactose medium containing X-gal. Blue colonies represent fragments that are associated with the indicated ω fragments. N-terminal fragments, which are associated with the mitochondrial ω fragments, display a functional MTS. Controls: 40 N-terminal as of KGD2α (mitochondrial), 40 N-terminal aas of HxKα (cytosolic), and Aco1α (dual-targeted). (**C**) α-complementation of *TDH* gene constructs. Yeast colonies expressing ωc or ωm were co-expressed with the α-fused *TDH1*, *TDH2*, and *TDH*3 wild-type genes (as described in (**A**), illustration 2); and with the α-fused constructs Δ*MTS-TDH2* and Δ*MTS-TDH3* (as described in (**A**), illustration 3), were submitted to α-complementation (as described in (**B**)). Cytosolic control was *ΔMTS-ACO1*α, and the mitochondrial control was *KGD2*α. (**D**) α-complementation of *TDH* mutants. *mut2p-TDH2α* and *mut5p-TDH2α* (described in (**A**), illustration 4); *FUMMTS-mut2p-TDH2α* and *FUMMTS-mut5p-TDH2α* are the corresponding mutants above, containing an upstream fused *FUM-MTS* (as described in (**A**), illustration 5), were submitted to α-complementation as described in (**B**). KGD2α is a mitochondrial control.

**Figure 5 cells-12-01550-f005:**
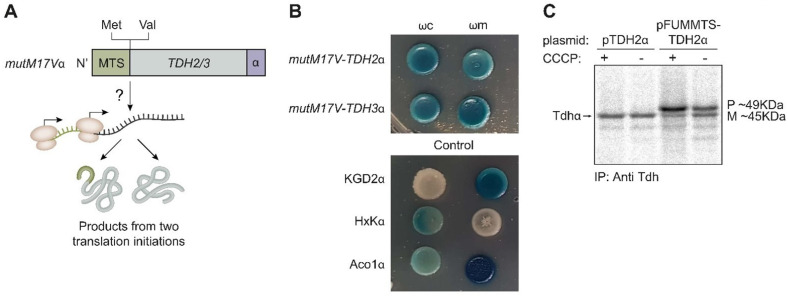
Tdh proteins display a single translation product. (**A**) Illustration of *TDH2/3* mutant (grey) in which the 17th amino acid methionine (Met) was replaced by valine (Val). This *mutM17Vα* removes a potential translation initiation codon, which could theoretically generate two translation products. (**B**) α-complementation of mutant proteins shows mitochondrial localization is preserved. (**C**) Yeast strains harboring the indicated plasmids were induced in galactose medium and labeled with [^35^S] methionine in the presence or in the absence of CCCP. Total cell extracts were prepared, immunoprecipitated with alpha antibody, and analyzed by SDS-PAGE and autoradiography. P/M on the right indicates precursor and mature forms of *FUMMTS-TDH2α*, and the arrow on the left indicates wild type Tdh*α*.

**Figure 6 cells-12-01550-f006:**
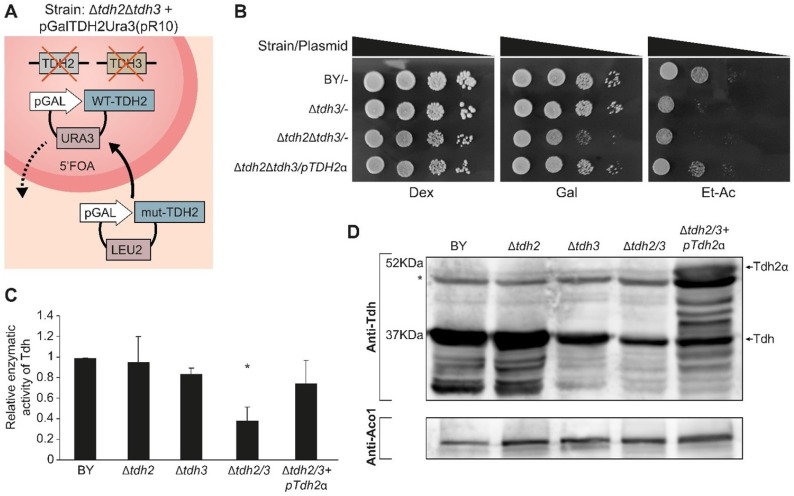
Generation of a Δ*tdh2*Δ*tdh3* double mutant and plasmid shuffling. (**A**) An illustration of Δ*tdh2*Δ*tdh3* + pGalTDH2Ura3(pR10) yeast strain; *TDH2* and *TDH3* genes were knocked out of the yeast genome in a strain harboring a Ura3 plasmid encoding *TDH2* under the GAL10 promoter, pGalTDH2-Ura3 (pR10). pR10 can be shuffled out under 5′FOA selection (broken arrow), and a plasmid with a Leu2 marker encoding a *TDH* gene or derivatives can be shuffled in (solid arrow). The latter can carry *TDH* mutants, which allow us to follow cell survival phenotypes. (**B**) The indicated yeast strains were grown overnight in selective medium, diluted to A600 = 0.4, and 5 µL of 10-fold serial dilutions were spotted onto Dex, dextrose; Gal, galactose; and Et-Ac, ethanol-acetate plates, which were incubated at 30 °C for 2–3 days. (**C**) The indicated yeast strains harboring plasmids were grown to mid-log phase, and cell extracts were generated and assayed for Tdh enzymatic activity at 340 nm using Glyceraldehyde 3-phosphate and NAD+ as substrates (mean ± SEM [*n* = 3], two-tailed Student’s *t*-test * *p* < 0.05). (**D**) Equivalent portions from each strain extract (from (**C**)) were analyzed by western blot with antibodies against Tdh. * marks nonspecific bands.

**Figure 7 cells-12-01550-f007:**
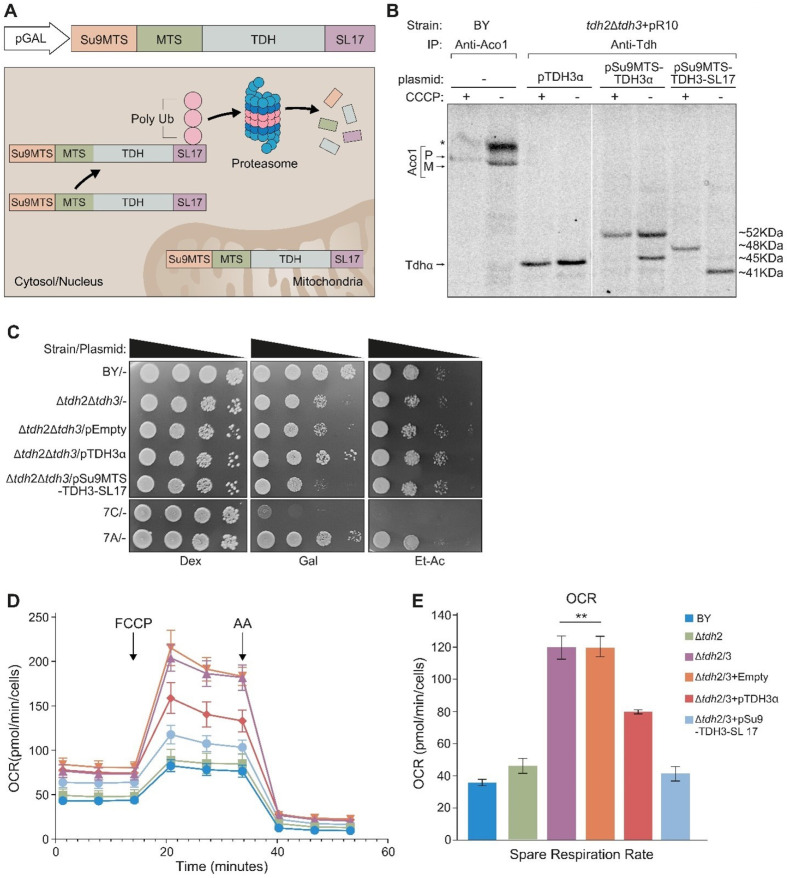
A Δ*tdh2*Δ*tdh3* strain OCR can be restored to wild-type values by Tdh expressed solely in mitochondria. (**A**) Illustration of the *Su9-Tdh3-SL17* construct, which encodes Tdh3 fused to the MTS of Su9 at its N-terminus and SL17 at its C-terminus. This Tdh3 derivative is expressed under GAL10 promoter and is present solely in mitochondria. The cytosolic derivatives are degraded by the ubiquitination-proteasomal system due to the degron tag (SL17), while the mitochondrial proteins are protected inside the organelle. (**B**) Yeast strains harboring the indicated plasmids were induced overnight in selective medium and labeled with [^35^S] methionine in the presence or in the absence of CCCP. Total cell extracts were prepared, immunoprecipitated with Tdh or Aco1 (control) antibodies, and analyzed by SDS-PAGE and autoradiography. Top and lower arrows on the left indicate the precursor (P) and mature form (M) of Aco1, respectively; The middle arrow indicates Tdhα derivative proteins, and the arrow on the lower left indicates Tdh untagged proteins; * nonspecific bands. (**C**) Yeast strains harboring the indicated plasmids were grown to A600 = 0.4, and 5 µL of 10-fold serial dilutions were spotted onto Dex, dextrose, Gal, galactose, and Et-Ac, ethanol-acetate medium. Plates were incubated at 30 °C for 2–3 days. (**D**) Yeast strains harboring plasmids were grown and diluted to A600 = 0.1 in ethanol-acetate seahorse buffer. Oxygen consumption rate (OCR) was measured every 3 min on poly-L-lysine-coated Seahorse XF96 microplates. FCCP and antimycin A (AA) were sequentially added to evaluate mitochondrial respiratory states. (**E**) The spare respiration capacity (maximal minus basal, Appendix A) OCR as obtained from three independent experiments (mean ± SEM [*n* = 3]), two-tailed Student’s *t*-test ** *p* < 0.001) described in (**D**).

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
