# Peer review of "Systematic Approaches to Study Eclipsed Targeting of Proteins Uncover a New Family of Mitochondrial Proteins"

_cells, 2023, doi:10.3390/cells12111550_

Round 1

Reviewer 1 Report

In their novel previous works, the Pines laboratory showed that phenomenon of dually distributed proteins is much more common than previously thought.  Their initial focus was abundant mitochondrial proteins that exhibit minor (eclipsed) distribution. In this work the conducted comprehensive search to identify soluble nonmitochondrial proteins that may eclipse distribute into mitochondria.  The methods utilized include:  genome wide screen, in yeast, using a complementation assay, bioinformatic analysis of proteins that may carry putative MTS, proteome analysis of mitochondria and more. The analysis yielded large number of soluble nonmitochondrial proteins with eclipsed distribution.  One of them, trios phosphate dehydrogenase (TDH), was further studied in depth.  It clearly shown that the mitochondrial distribution of TDH is required for the function of mitochondria, when yeast grown in carbon source that require functional mitochondria.  This work uncovers a previously unknown mitochondrial localization which affect the function of mitochondria.

Author Response

Thanks for the reviewer's comments

Reviewer 2 Report

Comments and Suggestions for Authors

Maayan Mark et al. investigate the phenomenon of dual localization of proteins. In particular, they focus on those proteins known to reside and have main activity in other cellular areas that are also localized -albeit at low levels- to mitochondria (eclipsed). The concept of this manuscript is general interest of this research field since it gives a major contribution to knowledge about mitochondrial eclipsed proteins as a result of a considerable body of work.

The manuscript brings together a large set of complementary data from challenging experiments and is presented clearly, although in some sections the extreme summary risks compromising clarity and makes it difficult to follow the complexity of the experiments and the data acquired.

The manuscript should be improved by addressing a couple of points and is accepted after minor revision.

General concept comments

1.     Fig. 3D-E — Comments on Western blots is not exhaustive. How do the authors explain the difference between the anti-Tdh signal (36KDa) shown in D and the anti-Tdh bands in E? The description of the results should be clearer and more complete in order to make the data easy to interpret and understand. 

2.     Are the pTdh1-6His, pTdh2-6His, pTdh3-6His enzymatically active? This would strengthen the data relating to the distribution pattern of the protein.

 3.     Fig. 3E Legend — The reference to (A) is incorrect and the legend should be more complete.

 4.     Materials and Methods section — The section on methods should be integrated, better specifying the details in order to make the manuscript’s results reproducible based on the details given. In many cases the protocol descriptions are too brief. For example, with regard to iTRAQ, SDS-PAGE and Western blot analysis (the quantities loaded on the SDS page are never specified) and enzymatic activity.

5.     Authors chose the TDH1/2/3 gene family as an example to prove that the eclipsed proteins are deliberately targeted to mitochondria and have a function there. Authors should comment on additional identified eclipsed mitochondrial proteins: families of proteins, particular functions, or any other aspect (if any) that allows to make a generic analysis.

Minor/Specific concerns: 

1.  Pag. 4 — line100/102 Check the color of Fig. 1B: orange and brown are not exactly the colors visible in the picture. 

2.     Pag. 5 — Paragraph 2.3: Check the spaces in the title

3.     Pag. 15 — Line 55: Check bold and line spacing

Authors please check for some inaccuracies in the English language (i.e. the adverb indeed is often repeated).

Author Response

General concept comments

  1. Fig. 3D-E — Comments on Western blots is not exhaustive. How do the authors explain the difference between the anti-Tdh signal (36KDa) shown in D and the anti-Tdh bands in E? The description of the results should be clearer and more complete in order to make the data easy to interpret and understand. 

Response: There is a very slight difference in gel migration of Tdh1/2/3 and the Tdhs tagged with 6 histidines, Tdh1/2/3-6his. Since Tdh1/2/3-6his are expressed under the strong GAL10 promoter the 6his versions are predominant (Fig 3E). The description of these figures has now been revised accordingly.

  1. Are the pTdh1-6His, pTdh2-6His, pTdh3-6His enzymatically active? This would strengthen the data relating to the distribution pattern of the protein.

Response: All the tagged versions of Tdh1/2/3 either with 6his or a are functional as shown for example in 6B and 7C. They appear to be enzymatically active as shown in Figure 7C. These data are described in the manuscript.

  1. Fig. 3E Legend — The reference to (A) is incorrect and the legend should be more complete.

Response: The reference to (A) has now been corrected to (D).

  1. Materials and Methods section — The section on methods should be integrated, better specifying the details in order to make the manuscript’s results reproducible based on the details given. In many cases the protocol descriptions are too brief. For example, with regard to iTRAQ, SDS-PAGE and Western blot analysis (the quantities loaded on the SDS page are never specified) and enzymatic activity.

Response: The Materials and Methods section has been modified in particular regarding the iTRAQ analysis in which a paragraph has been added.

  1. Authors chose the TDH1/2/3 gene family as an example to prove that the eclipsed proteins are deliberately targeted to mitochondria and have a function there. Authors should comment on additional identified eclipsed mitochondrial proteins: families of proteins, particular functions, or any other aspect (if any) that allows to make a generic analysis.

Response: We have recently published a paper in Cells (Zhang Y, Karmon O, Das K, Wiener R, Lehming N, Pines O. 2022. Ubiquitination occurs in the mitochondrial matrix by eclipsed targeted components of the ubiquitin machinery. Cells. 11(24): 4109.) which is referred to in more detail in the discussion.

 Minor/Specific concerns: 

  1. Pag. 4 — line100/102 Check the color of Fig. 1B: orange and brown are not exactly the colors visible in the picture.

Response: corrected

  1. Pag. 5 — Paragraph 2.3: Check the spaces in the title

Response: corrected

  1. Pag. 15 — Line 55: Check bold and line spacing

Response: do not detect the problem.

Reviewer 3 Report

The study identifies proteins that are partially localised to the mitochondria. A genomic SWAT library was used to demonstrate potential partial mitochondrial localisations. The authors identify 143 novel mitochondrial matrix proteins. They show that dual localised proteins are more highly conserved. 

The authors then concentrate on the triosephosphate dehydrogenase proteins which possess potential mitochondrial targeting signals. The authors show that distribution of the proteins is dependent upon the MTS and is not due to translation from an alternative methionine. Growth on respritory medium depends on the presence of a TDH gene. The authors show that localisation of the TDH protein to the mitochondria rescues growth on respiritory medium.

The paper is well written and the results are of interest to a wide audience. The experiments are well conceived and the results interpreted appropriately. I reccommend the paper for publication.

Author Response

Thanks for the reviewer's comments

Reviewer 4 Report

The manuscript by the groups of Ophry Pines and Maya Schuldiner presents a systematic approach to identify proteins, which are typically localized outside mitochondria, but reveal a minor population inside mitochondria. Such proteins are termed eclipsed proteins. The authors used an elegant genetic screen to identify such eclipsed proteins. In addition, they used bioinformatic and proteomic strategies to define the set of eclipsed proteins. Following this strategy, they identified more than 140 proteins that localized into the mitochondrial matrix. The presented data are of high quality and interesting for a broad readership. This finding represents a valuable asset for further studies on mitochondrial function. Some minor controls are required to substantiate the localization of the presented examples triose-phosphate dehydrogenases.

1.     The authors present an impressive list of more than 140 eclipsed proteins. They should categorize these proteins using GO terms. Furthermore, have these proteins been identified in systematic proteomic approaches (e.g. Morgenstern et al., 2017 Cell Rep.)? In this context, the described proteomic data should be shown in the manuscript.

2.     The authors should provide an overview figure showing the workflow that leads to the identification of the eclipsed proteins.

3.     Figure 3E: the localization of the TDH proteins is not clear. There is only a small fraction of TDH proteins in the mitochondrial fraction. The authors should study whether this fraction is resistant to externally added proteinase K to confirm there localization inside mitochondria.

4.     Figures 7C: the rescue of the growth defect of tdh2 tdh3 double deletion strain by pSu9-MTS-TDH3 is hardly visible. Additional data should be provided to show some data to the functional relevance of the mitochondrial population.
